# Occam's razor is insufficient to infer the preferences of irrational agents

**Sören Mindermann** * [†]
Vector Institute
University of Toronto
soeren.mindermann@gmail.com

**Stuart Armstrong** * [‡]
Future of Humanity Institute
University of Oxford
stuart.armstrong@philosophy.ox.ac.uk

## Abstract

Inverse reinforcement learning (IRL) attempts to infer human rewards or preferences from observed behavior. Since human planning systematically deviates from rationality, several approaches have been tried to account for specific human shortcomings. However, the general problem of inferring the reward function of an agent of unknown rationality has received little attention. Unlike the well-known ambiguity problems in IRL, this one is practically relevant but cannot be resolved by observing the agent's policy in enough environments. This paper shows (1) that a No Free Lunch result implies it is impossible to uniquely decompose a policy into a planning algorithm and reward function, and (2) that even with a reasonable simplicity prior/Occam's razor on the set of decompositions, we cannot distinguish between the true decomposition and others that lead to high regret. To address this, we need simple 'normative' assumptions, which cannot be deduced exclusively from observations.

## 1  Introduction

In today's reinforcement learning systems, a simple reward function is often hand-crafted, and still sometimes leads to undesired behaviors on the part of RL agent, as the reward function is not well aligned with the operator's true goals[4]. As AI systems become more powerful and autonomous, these failures will become more frequent and grave as RL agents exceed human performance, operate at time-scales that forbid constant oversight, and are given increasingly complex tasks — from driving cars to planning cities to eventually evaluating policies or helping run companies. Ensuring that the agents behave in alignment with human values is known, appropriately, as the *value alignment problem* [Amodei et al., 2016, Hadfield-Menell et al., 2016, Russell et al., 2015, Bostrom, 2014, Leike et al., 2017].

One way of resolving this problem is to infer the correct reward function by observing human behaviour. This is known as Inverse reinforcement learning (IRL) [Ng and Russell, 2000, Abbeel and Ng, 2004, Ziebart et al., 2008]. Often, learning a reward function is preferred over imitating a policy: when the agent must outperform humans, transfer to new environments, or be interpretable. The reward function is also usually a (much) more succinct and robust task representation than the policy, especially in planning tasks [Abbeel and Ng, 2004]. Moreover, supervised learning of long-range and goal-directed behavior is often difficult without the reward function [Ratliff et al., 2006].

---

[*] Equal contribution.
[†] Work performed at Future of Humanity Institute.
[‡] Further affiliation: Machine Intelligence Research Institute, Berkeley, USA.
[4] See for example the game CoastRunners, where an RL agent didn't finish the course, but instead found a bug allowing it to get a high score by crashing round in circles https://blog.openai.com/faulty-reward-functions/.

Usually, the reward function is inferred based on the assumption that human behavior is optimal or noisily optimal. However, it is well-known that humans deviate from rationality in *systematic*, non-random ways [Tversky and Kahneman, 1975]. This can be due to specific biases such as time-inconsistency, loss aversion and hundreds of others, but also limited cognitive capacity, which leads to forgetfulness, limited planning and false beliefs. This limits the use of IRL methods for tasks that humans don't find trivial.

Some IRL approaches address specific biases [Evans et al., 2015b,a], and others assume noisy rationality [Ziebart et al., 2008, Boularias et al., 2011]. But a general framework for inferring the reward function from suboptimal behavior does not exist to our knowledge. Such a framework needs to infer two unobserved variables simultaneously: the human reward function and their planning algorithm[5] which connects the reward function with behaviour, henceforth called a *planner*.

The task of observing human behaviour (or the human policy) and inferring from it the human reward function and planner will be termed *decomposing* the human policy. This paper will show there is a No Free Lunch theorem in this area: it is impossible to get a unique decomposition of human policy and hence get a unique human reward function. Indeed, *any* reward function is possible. And hence, if an IRL agent acts on what it believes is the human policy, the potential regret is near-maximal. This is another form of unidentifiability of the reward function, beyond the well-known ones [Ng and Russell, 2000, Amin and Singh, 2016].

The main result of this paper is that, unlike other No Free Lunch theorems, this unidentifiability does not disappear when regularising with a general simplicity prior that formalizes Occam's razor [Vitanyi and Li, 1997]. This result will be shown in two steps: first, that the simplest decompositions include degenerate ones, and secondly, that the most 'reasonable' decompositions according to human judgement are of high complexity.

So, although current IRL methods can perform well on many well-specified problems, they are fundamentally and philosophically incapable of establishing a 'reasonable' reward function for the human, no matter how powerful they become. In order to do this, they will need to build in 'normative assumptions': key assumptions about the reward function and/or planner, that cannot be deduced from observations, and allow the algorithm to focus on good ways of decomposing the human policy.

Future work will sketch out some potential normative assumptions that can be used in this area, making use of the fact that humans assess each other to be irrational, and often these assessments agree. In view of the No Free Lunch result, this shows that humans must share normative assumptions.

One of these 'normative assumption' approaches is briefly illustrated in an appendix, while another appendix demonstrates how to use the planner-reward formalism to define when an agent might be manipulating or overriding human preferences. This happens when the agent pushes the human towards situations where their policy is very suboptimal according to their reward function.

## 2   Related Work

In the first IRL papers from Ng and Russell [2000] and Abbeel and Ng [2004] a max-margin algorithm was used to find the reward function under which the observed policy most outperforms other policies. Suboptimal behavior was first addressed explicitly by Ratliff et al. [2006] who added slack variables to allow for suboptimal behavior. This finds reward functions such that the observed policy outperforms most other policies and the biggest margin by which another policy outperforms it is minimal, i.e. the observed policy has low regret. Shiarlis et al. [2017] introduce a modern max-margin technique with an approximate planner in the optimisation.

However, the max-margin approach has mostly been replaced by the max entropy IRL [Ziebart et al., 2008]. Here, the assumption is that observed actions or trajectories are chosen with probability proportional to the exponent of their value. This assumes a specific suboptimal planning algorithm which is *noisily* rational (also known as *Boltzmann*-rational). Noisy rationality explains human behavior on various data sets better [Hula et al., 2015]. However, Evans et al. [2015b] and Evans et al. [2015a] showed that this can fail since humans deviate from rationality in systematic, non-random ways. If noisy rationality is assumed, repeated suboptimal actions throw off the inference.

Literature on inferring the reasoning capabilities of an agent is scarce. Evans et al. [2015b] and Evans et al. [2015a] use Bayesian inference to identify specific planning biases such as myopic planning and hyperbolic time-discounting. They simultaneously infer the agent's preferences. Cundy and Filan [2018] adds bias resulting from hierarchical planning. Hula et al. [2015] similarly let agents infer features of their opponent's reasoning such as planning depth and impulsivity in simple economic games. Recent work learns the planning algorithm with two assumptions: being close to noisily rational in a high-dimensional planner space and supervised planner-learning [Anonymous, 2019].

The related ideas of meta-reasoning [Russell, 2016], computational rationality [Lewis et al., 2014] and resource rationality [Griffiths et al., 2015] may create the possibility to redefine irrational behavior as rational in an 'ancestral' distribution of environments where the agent optimises its rewards by choosing among the limited computations it is able to perform or jointly minimising the cost of computation and maximising reward. This could in theory redefine many biases as computationally optimal in some distribution of environments and provide priors on human planning algorithms. Unfortunately the problem of doing this in practice seems to be extremely difficult — and it assumes that human goals are roughly the same as evolution's goals, which is certainly not the case.

## 3  Problem setup and background

A human will be performing a series of actions, and from these, an agent will attempt to estimate both the human's reward function and their planning algorithm.

The environment $M$ in which the human operates is an MDP/R, a Markov Decision Process without reward function (a *world-model* [Hadfield-Menell et al., 2017]). An MDP/R is defined as a tuple, $\langle \mathcal{S}, \mathcal{A}, T, \hat{s} \rangle$ consisting of a discrete state space $\mathcal{S}$, a finite action space $\mathcal{A}$, a fixed starting state $\hat{s}$, and a probabilistic transition function $T : \mathcal{S} \times \mathcal{A} \times \mathcal{S} \to [0, 1]$ to the next state (also called the *dynamics*). At each step, the human is in a certain state $s$, takes a certain action $a$, and ends up in a new state $s'$ as given by $T(s' \mid s, a)$.

Let $\mathcal{R} = \{R : \mathcal{S} \times \mathcal{A} \to [-1, 1]\} = [-1, 1]^{\mathcal{S} \times \mathcal{A}}$ be the space of candidate reward functions; a given $R$ will map any state-reward pair to a reward value in the interval $[-1, 1]$.

Let $\Pi$ be the space of deterministic, Markovian policies. So $\Pi$ is the space of functions $\mathcal{S} \to \mathcal{A}$. The human will be following the policy $\dot{\pi} \in \Pi$.

The results of this paper apply to both discounted rewards and episodic environments settings[6].

### 3.1  Planners and reward functions: decomposing the policy

The human has their reward function, and then follows a policy that presumably attempts to maximise it. Therefore there is something that bridges between the reward function and the policy: a piece of greater or lesser rationality that transforms knowledge of the reward function into a plan of action.

This bridge will be modeled as a *planner* $p : \mathcal{R} \to \Pi$, a function that takes a reward and outputs a policy. This planner encodes all the rationality, irrationality, and biases of the human. Let $\mathcal{P}$ be the set of planners. The human is therefore defined by a *planner-reward* pair $(p, R) \in \mathcal{P} \times \mathcal{R}$. Similarly, $(p, R)$ with $p(R) = \pi$ is a *decomposition* of the policy $\pi$. The task of the agent is to find a 'good' decomposition of the human policy $\dot{\pi}$.

### 3.2  Compatible pairs and evidence

The agent can observe the human's behaviour and infer their policy from that. In order to simplify the problem and separate out the effect of the agent's learning, we will assume the agent has perfect knowledge of the human policy $\dot{\pi}$ and of the environment $M$. At this point, the agent cannot learn anything by observing the human's actions, as it can already perfectly predict these.

Then a pair $(p, R)$ is defined to be *compatible* with $\dot{\pi}$, if $p(R) = \dot{\pi}$ — thus that pair is a possible candidate for decomposing the human policy into the human's planner and reward function.

# 4 Irrationality-based unidentifiability

Unidentifiability of the reward is a well-known problem in IRL [Ng and Russell, 2000]. Amin and Singh [2016] categorise the problem into *representational* and *experimental* unidentifiability. The former means that adding a constant to a reward function or multiplying it with a positive scalar does not change what is optimal behavior. This is unproblematic as rescaling the reward function doesn't change the preference ordering. The latter can be resolved by observing optimal policies in a whole class of MDPs which contains all possible transition dynamics. We complete this framework with a third kind of identifiability, which arises when we observe suboptimal agents. This kind of unidentifiability is worse as it cannot necessarily be resolved by observing the agent in many tasks. In fact, it can lead to almost arbitrary regret.

## 4.1 Weak No Free Lunch: unidentifiable reward function and half-maximal regret

The results in this section show that without assumptions about the rationality of the human, all attempts to optimise their reward function are essentially futile. Everitt et al. [2017] work in a similar setting as we do: in their case, a corrupted version of the reward function is observed. The problem our case is that a 'corrupted' version $\dot{\pi}$ of an optimal policy $\pi_{\dot{R}}^*$ is observed and used as information to optimise for the ideal reward $\dot{R}$. A No Free Lunch result analogous to theirs applies in our case; both resemble the No Free Lunch theorems for optimisation [Wolpert and Macready, 1997].

More philosophically, this result is as an instance of the well-known *is-ought* problem from meta-ethics. Hume [1888] argued that what *ought* to be (here, the human's reward function) can never be concluded from what *is* (here, behavior) without extra assumptions. Equivalently, the human reward function cannot be inferred from behavior without assumptions about the planning algorithm $p$. In probabilistic terms, the likelihood $P(\pi|R) = \sum_{p \in \mathcal{P}} P(\pi \mid R, p) P(p)$ is undefined without $P(p)$. As shown in Section 5 and Section 5.2, even a simplicity prior on $p$ and $R$ will not help.

### 4.1.1 Unidentifiable reward functions

Firstly, we note that compatibility ($p(R) = \dot{\pi}$), puts no restriction on $R$, and few restrictions on $p$:
**Theorem 1.** *For all $\pi \in \Pi$ and $R \in \mathcal{R}$, there exists a $p \in \mathcal{P}$ such that $p(R) = \pi$.*

*For all $p \in \mathcal{P}$ and $\pi \in \Pi$ in the image of $p$, there exists an $R$ such that $p(R) = \pi$.*

*Proof.* Trivial proof: define the planner[7] $p$ as mapping all of $\mathcal{R}$ to $\pi$; then $p(R) = \pi$. The second statement is even more trivial, as $\pi$ is in the image of $p$, so there must exist $R$ with $p(R) = \pi$. □

### 4.1.2 Half-maximal regret

The above shows that the reward function cannot be constrained by observation of the human, but what about the expected long-term value? Suppose that an agent is unsure what the actual human reward function is; if the agent itself is acting in an MDP/R, can it follow a policy that minimises the possible downside of its ignorance?

This is prevented by a recent No Free Lunch theorem. Being ignorant of the reward function one should maximise is equivalent of having a *corrupted reward channel* with arbitrary corruption. In that case, Everitt et al. [2017] demonstrated that whatever policy $\pi$ the agent follows, there is a $R \in \mathcal{R}$ for which $\pi$ is half as bad as the worst policy the agent could have followed. Specifically, let $V_R^\pi(s)$ be the expected return of reward function $R$ from state $s$, given that the agent follows policy $\pi$. If $\pi$ was the optimal policy for $R$, then this can be written as $V_R^*(s)$. The regret of $\pi$ for $R$ at $s$ is given by the difference:
$$\text{Reg}(\pi, R)(s) = V_R^*(s) - V_R^\pi(s).$$
Then Everitt et al. [2017] demonstrates that for any $\pi$,
$$\max_{R \in \mathcal{R}} \text{Reg}(\pi, R)(s) \geq \frac{1}{2} \left( \max_{\pi' \in \Pi, R \in \mathcal{R}} \text{Reg}(\pi', R)(s) \right).$$
So for any compatible $(p, R) = \dot{\pi}$, we cannot rule out that maximizing $R$ leads to at least half of the worst-case regret.

# 5 Simplicity of degenerate decompositions

Like many No Free Lunch theorems, the result of the previous section is not surprising given there are no assumptions about the planning algorithm. No Free Lunch results are generally avoided by placing a simplicity prior on the algorithm, dataset, function class or other object [Everitt et al., 2014]. This amounts to saying algorithms can benefit from regularisation. This section is dedicated to showing that, surprisingly, simplicity does not solve the No Free Lunch result.

Our simplicity measure is minimum description length of an object, defined as Kolmogorov complexity [Kolmogorov, 1965], the length of the shortest program that outputs a string describing the object. This is the most general formalization of Occam's razor we know of [Vitanyi and Li, 1997]. Appendix A explores how the results extend to other measures of complexity, such as those that include computation time. We start with informal versions of our main results.

**Theorem 2** (Informal simplicity theorem)**.** *Let $(\dot{p}, \dot{R})$ be a 'reasonable' planner-reward pair that captures our judgements about the biases and rationality of a human with policy $\dot{\pi} = \dot{p}(\dot{R})$. Then there are degenerate planner-reward pairs, compatible with $\dot{\pi}$, of lower complexity than $(\dot{p}, \dot{R})$, and a pair $(\dot{p}', -\dot{R})$ of similar complexity to $(\dot{p}, \dot{R})$, but with opposite reward function.*

There are a few issues with this theorem as it stands. Firstly, simplicity in algorithmic information theory is relative to the computer language (or equivalently Universal Turing Machine) $L$ used [Ming and Vitányi, 2014, Calude, 2002], and there exists languages in which the theorem is clearly false: one could choose a degenerate language in which $(\dot{p}, \dot{R})$ is encoded by the string '0', for example, and all other planner-reward pairs are of extremely long length. What constitutes a 'reasonable' language is a long-standing open problem, see Leike et al. [2017] and Müller [2010]. For any pair of languages, complexities differ only by a constant, the amount required for one language to describe the other, but this constant can be arbitrarily large.

Nevertheless, this section will provide grounds for the following two semi-formal results:

**Proposition 3.** *If $\dot{\pi}$ is a human policy, and $L$ is a 'reasonable' computer language, then there exists degenerate planner-reward pairs amongst the pairs of lowest complexity compatible with $\dot{\pi}$.*

**Proposition 4.** *If $\dot{\pi}$ is a human policy, and $L$ is a 'reasonable' computer language with $(\dot{p}, \dot{R})$ a compatible planner-reward pair, then there exist a pair $(\dot{p}', -\dot{R})$ of comparable complexity to $(\dot{p}, \dot{R})$, but opposite reward function.*

The last part of Theorem 2, the fact that any 'reasonable' $(\dot{p}, \dot{R})$ is expected to be of higher complexity, will be addressed in Section 6.

## 5.1 Simple degenerate pairs

The argument in this subsection will be that 1) the complexity of $\dot{\pi}$ is close to a lower bound on any pair compatible with it and 2) degenerate decompositions are themselves close to this bound. The first statement follows because for any decomposition $(p, R)$ compatible with $\dot{\pi}$, the map $(p, R) \mapsto p(R) = \dot{\pi}$ will be a simple one, adding little complexity. And if a compatible pair $(p', R')$ can be from $\dot{\pi}$ with little extra complexity, then it too will have a complexity close to the minimal complexity of any other pair compatible with it. Therefore we will first produce three degenerate pairs that can be simply constructed from $\dot{\pi}$.

### 5.1.1 The degenerate pairs

We can define the trivial constant reward function 0, and the greedy planner $p_g$. The greedy planner $p_g$ acts by taking the action that maximises the immediate reward in the current state and the next action. Thus[8] $p_g(R)(s) = \operatorname{argmax}_a R(s, a)$. We can also define the anti-greedy planner $-p_g$, with $-p_g(R)(s) = \operatorname{argmin}_a R(s, a)$. In general, it will be useful to define the negative of a planner:

**Definition 5.** If $p : \mathcal{R} \to \Pi$ is a planner, the planner $-p$ is defined by $-p(R) = p(-R)$.

For any given policy $\pi$, we can define the *indifferent* planner $p_\pi$, which maps any reward function to $\pi$. We can also define the reward function $R_\pi$, so that $R_\pi(s, a) = 1$ if $\pi(s) = a$, and $R_\pi(s, a) = 0$ otherwise. The reward function $-R_\pi$ is defined to be the negative of $R_\pi$. Then:

**Lemma 6.** *The pairs $(p_\pi, 0)$, $(p_g, R_\pi)$, and $(-p_g, -R_\pi)$ are all compatible with $\pi$.*

*Proof.* Since the image $p_\pi$ is $\pi$, $p_\pi(0) = \pi$. Now, $R_\pi(s, a) > 0$ iff $\pi(s) = a$, hence for all $s$:

$$p_g(R_\pi)(s) = \underset{a}{\operatorname{argmax}} R_\pi(s, a) = \pi(s),$$

so $p_g(R_\pi) = \pi$. Then $-p_g(-R_\pi) = p_g(-(-R_\pi)) = p_g(R_\pi) = \pi$, by Definition 5. $\qquad\square$

### 5.1.2 Complexity of basic operations

We will look the operations that build the degenerate planner-reward pairs from any compatible pair:

1. For any planner $p$, $f_1(p) = (p, 0)$ as a planner-reward pair.
2. For any reward function $R$, $f_2(R) = (p_g, R)$.
3. For any planner-reward pair $(p, R)$, $f_3(p, R) = p(R)$.
4. For any planner-reward pair $(p, R)$, $f_4(p, R) = (-p, -R)$.
5. For any policy $\pi$, $f_5(\pi) = p_\pi$.
6. For any policy $\pi$, $f_6(\pi) = R_\pi$.

These will be called the basic operations, and there are strong arguments that reasonable computer languages should be able to express them with short programs. The operation $f_1$, for instance, is simply appending the flat trivial 0, $f_2$ appends a planner defined by the simple[9] search operator $\operatorname{argmax}$, $f_3$ applies a planner to the object — a reward function — that the planner naturally acts on, $f_4$ is a double negation, while $f_5$ and $f_6$ are simply described in subsubsection 5.1.1.

From these basic operations, we can define three composite operations that map any compatible planner-reward pair to one of the degenerate pairs (the element $F_4 = f_4$ is useful for later definitions). Thus define

$$F = \{F_1 = f_1 \circ f_5 \circ f_3, \; F_2 = f_2 \circ f_6 \circ f_3, \; F_3 = f_4 \circ f_2 \circ f_6 \circ f_3, \; F_4 = f_4\}.$$

For any $\dot\pi$-compatible pair $(p, R)$ we have $F_1(p, R) = (p_{\dot\pi}, 0)$, $F_2(p, R) = (p_g, R_{\dot\pi})$, and $F_3(p, R) = (-p_g, -R_{\dot\pi})$ (see the proof of Proposition 7).

Let $K_L$ denote Kolmogorov complexity in the language L: the shortest algorithm in $L$ that generates a particular object. We define the $F$-complexity of $L$ as

$$\max_{(p,R), F_i \in F} K_L(F_i(p, R)) - K_L(p, R).$$

Thus the $F$-complexity of $L$ is how much the $F_i$ potentially increase[10] the complexity of pairs.

For a constant $c \geq 0$, this allows us to formalise what we mean by $L$ being a $c$-reasonable language for $F$: that the $F$-complexity of $L$ is at most $c$. A reasonable language is a $c$-reasonable language for a $c$ that we feel is intuitively low enough.

### 5.1.3 Low complexity of degenerate planner-reward pairs

To formalise the concepts 'of lowest complexity', and 'of comparable complexity', choose a constant $c \geq 0$, then $(p, R)$ and $(p', R')$ are of 'comparable complexity' if

$$||K_L(p, R) - K_L(p', R')|| \leq c.$$

For a set $S \subset \mathcal{P} \times \mathcal{R}$, the pair $(p, R) \in S$ is amongst the lowest complexity in $S$ if

$$||K_L(p, R) - \min_{(p',R') \in S} K_L(p', R')|| \leq c,$$

thus $K_L$ is within distance $c$ of the minimum complexity element of $S$. Now formalize Proposition 3:

**Proposition 7.** *If $\dot{\pi}$ is the human policy, $c$ defines a reasonable measure of comparable complexity, and $L$ is a $c$-reasonable language for $F$, then the degenerate planner-reward pairs $(p_{\dot{\pi}}, 0)$, $(p_g, R_{\dot{\pi}})$, and $(-p_g, -R_{\dot{\pi}})$ are amongst the pairs of lowest complexity among the pairs compatible with $\dot{\pi}$.*

*Proof.* By Lemma 6, $(p_{\dot{\pi}}, 0)$, $(p_g, R_{\dot{\pi}})$, and $(-p_g, -R_{\dot{\pi}})$ are compatible with $\dot{\pi}$. By the definitions of the $f_i$ and $F_i$, for $(p, R)$ compatible with $\dot{\pi}$, $f_3((p, R)) = p(R) = \dot{\pi}$ and hence

$$F_1(p, R) = f_1 \circ f_5(\dot{\pi}) = f_1(p_{\dot{\pi}}) = (p_{\dot{\pi}}, 0),$$
$$F_2(p, R) = f_2 \circ f_6(\dot{\pi}) = f_2(R_{\dot{\pi}}) = (p_g, R_{\dot{\pi}}),$$
$$F_3(p, R) = f_4 \circ F_2(p, R) = (-p_g, -R_{\dot{\pi}}).$$

Now pick $(p, R)$ to be the simplest pair compatible with $\dot{\pi}$. Since $L$ is $c$-reasonable for $F$, $K_L(p_{\dot{\pi}}, 0) \leq c + K_L(p, R)$. Hence $(p_{\dot{\pi}}, 0)$ is of lowest complexity among the pairs compatible with $\dot{\pi}$; the same argument applies for the other two degenerate pairs. □

### 5.2 Negative reward

If $(\dot{p}, \dot{R})$ is compatible with $\dot{\pi}$, then so is $(-\dot{p}, -\dot{R}) = f_4(\dot{p}, \dot{R}) = F_4(\dot{p}, \dot{R})$. This immediately implies the formalisation of Proposition 4:

**Proposition 8.** *If $\dot{\pi}$ is a human policy, $c$ defines a reasonable measure of comparable complexity, $L$ is a $c$-reasonable language for $F$, and $(\dot{p}, \dot{R})$ is compatible with $\dot{\pi}$, then $(-\dot{p}, -\dot{R})$ is of comparable complexity to $(\dot{p}, \dot{R})$.*

So complexity fails to distinguish between a reasonable human reward function and its negative.

## 6 The high complexity of the genuine human reward function

Section 5 demonstrated that there are degenerate planner-reward pairs close to the minimum complexity among all pairs compatible with $\dot{\pi}$. This section will argue that any reasonable pair $(\dot{p}, \dot{R})$ is unlikely to be close to this minimum, and is therefore of higher complexity than the degenerate pairs. Unlike simplicity, reasonable decomposition cannot easily be formalised. Indeed, a formalization would likely already solve the problem, yielding an algorithm to maximize it. Therefore, the arguments in this section are mostly qualitative.

We use reasonable to mean 'compatible with human judgements about rationality'. Since we do not have direct access to such a decomposition, the complexity argument will be about showing the complexity of these human judgements. This argument will proceed in three stages:

1. Any reasonable $(\dot{p}, \dot{R})$ is of high complexity, higher than it may intuitively seem to us.

2. Even given $\dot{\pi}$, any reasonable $(\dot{p}, \dot{R})$ involves a high number of contingent choices. Hence any given $(\dot{p}, \dot{R})$ has high information (and thus high complexity), even given $\dot{\pi}$.

3. Past failures to find a simple $(\dot{p}, \dot{R})$ derived from $\dot{\pi}$ are evidence that this is tricky.

### 6.1 The complexity of human (ir)rationality

Humans make noisy and biased decisions all the time. Though noise is important [Kahneman et al., 2016], many biases, such as anchoring bias, overconfidence, planning fallacies, and so on, affect humans in a highly systematic way; see Kahneman and Egan [2011] for many examples.

Many people may feel that they have a good understanding of rationality, and therefore assume that assessing the (ir)rationality of any particular decision is not a complicated process. But an intuition for bias does not translate into a process for establishing a $(\dot{p}, \dot{R})$.

Consider the anchoring bias defined in Ariely et al. [2004], where irrelevant information — the last digits of social security numbers — changed how much people were willing to pay for goods. When defining a reasonable $(\dot{p}, \dot{R})$, it does not suffice to be aware of the existence of anchoring bias[11], but

one has to precisely quantify the extent of the bias — why does anchoring bias seem to be stronger for chocolate than for wine, for instance? And why these precise percentages and correlations, and not others? And can people's judgment tell which people are more or less susceptible to anchoring bias? And can one quantify the bias for a single individual, rather than over a sample?

Any given $(\dot{p}, \dot{R})$ can quantify the form and extent of these biases by computing objects like the regret function $\text{Reg}(\dot{p}, \dot{R})(s) := \text{Reg}(\dot{p}(\dot{R}), \dot{R})(s) = V_{\dot{R}}^*(s) - V_{\dot{R}}^{\dot{p}(\dot{R})}(s)$, which measures the divergence between the expected value of the actual and optimal human policies[12]. Thus any given $(\dot{p}, \dot{R})$ — which contains the information to compute quantities like $\text{Reg}(\dot{p}, \dot{R})(s)$ or similar measures of bias[13], in every state — carries a high amount of numerical information about bias, and hence a high complexity.

Since humans do not easily have access to this information, this implies that human judgement of irrationality is subject to Moravec's paradox [Moravec, 1988]. It is similar to, for example, social skills: though it seems intuitively simple to us, it is highly complex to define in algorithmic terms.

Other authors have argued directly for the complexity of human values, from fields as diverse as computer science, philosophy, neuroscience, and economics [Minsky, 1984, Bostrom, 2014, Glimcher et al., 2009, Muehlhauser and Helm, 2012, Yudkowsky, 2011].

## 6.2 The contingency of human judgement

The previous section showed that reasonable $(\dot{p}, \dot{R})$ carry large amounts of information/complexity, but the key question is whether it requires information *additional* to that in $\dot{\pi}$. This section will show that even when $\dot{\pi}$ is known, there are many contingent choices that need to be made to define any specific reasonable $(\dot{p}, \dot{R})$. Hence any given $(\dot{p}, \dot{R})$ contains a large amount of information beyond that in $\dot{\pi}$, and hence is of higher complexity.

Reasons to believe that human judgement about reasonable $(\dot{p}, \dot{R})$ contains many contingent choices:

- There is a variability of human judgement between cultures. When Miller [1984] compared American and Indian assessments of the same behaviours, they found systematically different explanations for them[14] Basic intuitions about rationality also vary between cultures [Nisbett et al., 2001, Brück, 1999].

- There is a variability of human judgement within a single culture. When Slovic and Tversky [1974] analysed the "Allais Paradox", they found that different people gave different answers as to what the rational behaviour was in their experiments.

- There is evidence of variability of human judgement within the same person. Slovic and Tversky [1974] further attempted to argue for the rationality of one of the answers. This sometimes resulted in the participant sometimes changing their minds, and contradicting their previous assessment of rationality.

- There is a variability of human judgement for the same person assessing their own values, caused by differences as trivial as question ordering [Schuman and Ludwig, 1983]. So human meta-judgement, of own values and rationality, is also contingent and variable.

- People have partial bias blind spots around their own biases [Scopelliti et al., 2015].

Thus if a human is following policy $\dot{\pi}$, a decomposition $(\dot{p}, \dot{R})$ would provide additional information about the cultural background of the decomposer, their personality within their culture, and even about the past history of the decomposer and how the issue is being presented to them. Those last pieces prevents us from 'simply' using the human's own assessment of their own rationality, as that assessment is subject to change and re-interpretation depending on their possible histories.

### 6.3 The search for human rationality models

One final argument that there is no simple algorithm for going from $\dot{\pi}$ to $(\dot{p}, \dot{R})$: many have tried and failed to find such an algorithm. Since the subject of human rationality has been a major one for several thousands of years, the ongoing failure is indicative — though not a proof — of the difficulties involved. There have been many suggested philosophical avenues for finding such a reward (such as reflective equilibrium [Rawls, 1971]), but all have been underdefined and disputed.

The economic concept of revealed preferences [Samuelson, 1948] is the most explicit, using the assumption of rational behaviour to derive human preferences. This is an often acceptable approximation, but can be taken too far: failure to take achieve an achievable goal does not imply that failure was desired. Even within the confines of economics, it has been criticised by behavioural economics approaches, such as prospect theory [Kahneman and Tversky, 2013] — and there are counter-criticisms to these.

Using machine learning to deduce the intentions and preferences of humans is in its infancy, but we can see non-trivial real-world examples, even in settings as simple as car-driving [Lazar et al., 2018].

Thus to date, neither humans nor machine learning have been able to find simple ways of going from $\dot{\pi}$ to $(\dot{p}, \dot{R})$, nor any simple and *explicit* theory for how such a decomposition could be achieved. This suggests that $(\dot{p}, \dot{R})$ is a complicated object, even if $\dot{\pi}$ is known. In conclusion:

**Conjecture 9** (Informal complexity proposition). If $\dot{\pi}$ is a human policy, and $L$ is a 'reasonable' computer language with $(\dot{p}, \dot{R})$ a 'reasonable' compatible planner-reward pair, then the complexity of $(\dot{p}, \dot{R})$ is not close to minimal amongst the pairs compatible with $\dot{\pi}$.

## 7 Conclusion

We have shown that some degenerate planner-reward decompositions of a human policy have near-minimal description length and argued that decompositions we would endorse do not. Hence, under the Kolmogorov-complexity simplicity prior, a formalization of Occam's Razor, the posterior would endorse degenerate solutions. Previous work has shown that noisy rationality is too strong an assumption as it does not account for bias; we tried the weaker assumption of simplicity, strong enough to avoid typical No Free Lunch results, but it is insufficient here.

This is no reason for despair: there is a large space to explore between these two extremes. Our hope is that with some minimal assumptions about planner and reward we can infer the rest with enough data. Staying close to agnostic is desirable in some settings: for example, a misspecified model of the human reward function can lead to disastrous decisions with high confidence [Milli et al., 2017]. Anonymous [2019] makes a promising first try — a high-dimensional parametric planner is initialized to noisy rationality and then adapts to fit the behavior of a systematically irrational agent.

How can we reconcile our results with the fact that humans routinely make judgments about the preferences and irrationality of others? And, that these judgments are often correlated from human to human? After all, No Free Lunch applies to human as well as artificial agents. Our result shows that they must be using shared priors, beyond simplicity, that are not learned from observations. We call these *normative assumptions* because they encode beliefs about which reward functions are more likely and what constitutes approximately rational behavior. Uncovering minimal normative assumptions would be an ideal way to build on this paper; Appendix C shows one possible approach.

## Acknowledgments.

We wish to thank Laurent Orseau, Xavier O'Rourke, Jan Leike, Shane Legg, Nick Bostrom, Owain Evans, Jelena Luketina, Tom Everrit, Jessica Taylor, Paul Christiano, Eliezer Yudkowsky, Stuart Russell, Dylan Hadfield-Menell, and Anders Sandberg, Adam Gleave, Rohin Shah, among many others. This work was supported by the Alexander Tamas programme on AI safety research, the Leverhulme Trust, and the Machine Intelligence Research Institute.

## Footnotes

[5] Technically we only need to infer the human reward function, but inferring that from behaviour requires some knowledge of the planning algorithm.

[6]The setting is only chosen for notational convenience: it also emulates discrete POMDPs, non-Markovianness (eg by encoding the whole history in the state) and pseudo-random policies.

[7]This is the 'indifferent' planner $p_\pi$ of subsubsection 5.1.1.

[8]Recall that $p_g$ is a planner, $p_g(R)$ is a policy, so $p_g(R)$ can be applied to states, and $p_g(R)(s)$ is an action.

[9] In most standard computer languages, $\operatorname{argmax}$ just requires a for-loop, a reference to $R$, a comparison with a previously stored value, and possibly the storage of a new value and the current action.

[10] $F$-complexity is non-negative: $F_4 \circ F_4$ is the identity, so that $K_L(F_4(p, R)) - K_L(p, R) = -(K_L(F_4(F_4(p, R)) - K_L(F_4(p, R)))$, meaning that $\max_{(p,R), F_4} K_L(F_4(p, R)) - K_F(p, R)$ must be non-negative; this is a reason to include $F_4$ in the definition of $F$.

[11] The fact that many cognitive biases have only been discovered recently argue against people having a good intuitive grasp of bias and rationality, as do people's persistent bias blind spots [Scopelliti et al., 2015].

[12]To exactly quantify the anchoring bias above, we could use a regret function that contrasts $\dot{\pi}$ with the same policy, but where the decision is optimal for one turn only (rather than for all turns, as in standard regret).

[13]In contrast, regret for the degenerate planner-reward pairs is trivial. $\text{Reg}(p_{\dot{\pi}}, 0)$ and $\text{Reg}(p_g, R_{\dot{\pi}})$ are identically zero — in the second case, since $p_g(R_{\dot{\pi}})$ is actually optimal for $R_{\dot{\pi}}$, getting the maximal possible reward — while $(-p_g, -R_{\dot{\pi}})$ has a regret that is identically $-1$ at each step.

[14]"Results show that there were cross-cultural and developmental differences related to contrasting cultural conceptions of the person [...] rather than from cognitive, experiential, and informational differences [...]."

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
