[Supplementary Material]

# A  Other measures of algorithmic complexity

It might be felt that Proposition 7 depends on using only the Kolmogorov/algorithmic complexity of $L$. For example, it seems that though the algorithm defining $R_\pi$ in subsubsection 5.1.1 is short, the running time of $(p_g, R_\pi)$ might be much longer than other compatible $(p, R)$ pairs. This is because $p_g$ defines an argmax over actions while $R_\pi(s, a)$ runs $\pi$ on $s$. Hence applying $p_g$ to $R_\pi$ requires running $\pi(s)$ as many times as $||\mathcal{A}||$, which is very inefficient.

We could instead use a measure of complexity that also uses the number of operations required to compute a pair [Schmidhuber, 2002].

For any object $S$, let $\alpha_S$ be an algorithm that generates $S$ as an output. If $S$ is a function that can be applied to another object $T$, then $\alpha_S(\alpha_T)$ generates $S(T)$ by generating $S$ with $\alpha_S$, whenever $S$ needs to look at $T$, it uses $\alpha_T$ to generate $T$.

For example, if $\alpha$ is an algorithm in the language of $L$, $l(\alpha)$ its length, and $t(\alpha)$ its running time, we could define the time-bounded Kolmogorov complexity,

$$Kt_L(p, R) = \min_{\alpha_p, \alpha_R} l(\alpha_p) + l(\alpha_R) + \log(t(\alpha_p(\alpha_R)))$$

$$KT_L(p, R) = \min_{\alpha_p, \alpha_R} l(\alpha_p) + l(\alpha_R) + t(\alpha_p(\alpha_R)).$$

The $Kt_L$ derives from Levin [1984], while $KT_L$ is closely related to the example in Allender [2001]. Note that instead of $l(\alpha_p) + l(\alpha_R)$ we could consider the length of a single algorithm that generates both $p$ and $R$; however, for the degenerate pairs we are considering, the length of such an algorithm is very close to $l(\alpha_p) + l(\alpha_R)$, as either $\alpha_p$ or $\alpha_R$ would be trivial.

The main result is that neither $Kt_L$ nor $KT_L$ complexity remove the No Free Lunch Theorem. For the degenerate pair $(p_{\dot\pi}, 0)$, nothing is gained, because its running time is comparable to $\dot\pi$. For the other two degenerate pairs, consider the situation where a planner takes as input not a reward function $R \in \mathcal{R}$, but the source code in $L$ of an algorithm that computes $R$. In that case, the previous proposition still applies:

**Proposition 10.** *The results of Proposition 7 still apply to $(p_{\dot\pi}, 0)$ if $Kt_L$ or $KT_L$ are used instead of $K_L$. If planner can take in algorithms generating reward functions, rather than simply reward functions, then the results of Proposition 7 still apply to $(p_g, R_{\dot\pi})$ and $(-p_g, -R_{\dot\pi})$ in this situation.*

*Proof.* The proof will only be briefly sketched. If $L$ is reasonable, $l(\alpha_0)$ can be very small (it's simply the zero function), and since $p_{\dot\pi}$ need not actually look at its input, $t(\alpha_{p_{\dot\pi}}(\alpha_0))$ can be simplified to $t(\alpha_{\dot\pi})$. Thus $Kt_L(p_{\dot\pi}, 0)$ and $KT_L(p_{\dot\pi}, 0)$ are close to the $Kt_L$ and $KT_L$ complexities of $\dot\pi$ itself.

For $(p_g, R_{\dot\pi})$, let $\alpha_{p_g}$ and $\alpha_{\dot\pi}$ be the algorithms that generates $p_g$ and $\dot\pi$ which are of lowest $Kt_L$-complexity.

Then define the algorithm $W(\alpha_{\dot\pi})$. This algorithm wraps $\alpha_{\dot\pi}$ up: first it takes inputs $s$ and $a$, then runs $\alpha_{\dot\pi}$ on $s$, then returns 1 if the output of that is $a$ and 0 otherwise. Thus $W(\alpha_{\dot\pi})$ is an algorithm for $R_{\dot\pi}$.

We also wrap $\alpha_{p_g}$ into $W'(\alpha_{p_g})$. Here, $W'(\alpha_{p_g})$, when provided with an input algorithm $\beta$, will check whether it is in the specific form $\beta = W(\alpha)$. If it is, it will run $\alpha$, and output its output. If it is not, it will run $\alpha_{p_g}$ on $\beta$.

If $L$ is reasonable, then $W(\alpha_{\dot\pi})$ is of length only slightly longer than $\alpha_{\dot\pi}$, and of runtime also only slightly longer, and the same goes for $W'(\alpha_{p_g})$ and $\alpha_{p_g}$ (indeed $W'(\alpha_{p_g})$ can have a shorter runtime than $\alpha_{p_g}$).

Now $W(\alpha_{\dot\pi})$ is an algorithm for $R_{\dot\pi}$, while $W'(\alpha_{p_g})$ always has the same output as $\alpha_{p_g}$. Notice that, when running the algorithm $W'(\alpha_{p_g})$ with $W(\alpha_{\dot\pi})$ as input, this is only slightly longer in both senses than simply running $\alpha_{\dot\pi}$: $W'(\alpha_{p_g})$ will analyse $W(\alpha_{\dot\pi})$, notice it is in the form $W$ of $\alpha_{\dot\pi}$, and then simply run $\alpha_{\dot\pi}$.

Thus the $Kt_L$ complexity of $(p_g, R_{\dot\pi})$ is only slightly higher than that of $\dot\pi$. The same goes for the $KT_L$ complexity, and for $(-p_g, -R_{\dot\pi})$. $\qquad\square$

Some other alternatives suggested have focused on bounding the complexity either of the reward function or the planner, rather than of both. This would clearly not help, as $(p_{\dot{\pi}}, 0)$ has a reward function of minimal complexity, while $(p_g, R_\pi)$ and $(-p_g, -R_\pi)$ have minimal complexity planner.

Some other ad-hoc ideas suggested that the complexity of the planner and the reward need to be comparable[15]. This would rule out the three standard degenerate solutions, but should allow others that spread complexity between planner and reward in whatever proportion is desired[16].

It seems that similar tricks could be performed with many other types of complexity measures. Thus simplicity of any form does not seem sufficient for resolving this No Free Lunch result.

# B  Overriding human reward functions

ML systems may, even today, influence humans by showing manipulative adds, and then naïvely concluding that the humans really like those products (since they then buy them). Even though the $(p, R)$ formalism was constructed to model rationality and reward function in a human, it turns out that it can also model situations where human preferences are overridden or modified.

That's because the policy $\dot{\pi}$ encodes the human action in all situations, including situations where they are manipulated or coerced. Therefore, overridden reward functions can be detected by divergence between $\dot{\pi}$ and a more optimal policy for the reward function $R$.

Manipulative ads are a very mild form of manipulation. More extreme versions could involve manipulative propaganda, drug injections or even coercive brain surgery — a form of human *wireheading* [Everitt and Hutter, 2016], where the agent changes the human's behaviour and apparent preferences. All these methods of manipulation[17] will be designated as the agent *overriding* the human reward function.

In the $(p, R)$ formalism, the reward function $R$ can be used to detect such overriding, distinguishing between legitimate optimisation (eg informative adds) and illegitimate manipulation/reward overriding (eg manipulative adds).

To model this, the agent needs to be able to act, so the setup needs to be extended. Let $M^*$ be the same MDP/R as $M$, except each state is augmented with an extra boolean variable: $\mathcal{S}^* = \mathcal{S} \times \{0, 1\}$. The extra boolean never changes, and its only effect is to change the human policy.

On $\mathcal{S}_0 = \mathcal{S} \times \{0\}$, the human follows $\dot{\pi}$; on $\mathcal{S}_1 = \mathcal{S} \times \{1\}$, the human follows an alternative policy $\pi^a = \pi^*_{R^a}$, which is defined as the policy that maximises the expectation of a reward function $R^a$.

The agent can choose actions from within the set $\mathcal{A}^a$. It can choose either $0$, in which case the human starts in $\hat{s}_0 = \hat{s} \times \{0\}$ without any override and standard policy $\dot{\pi}$. Or it can choose $(1, R^a)$, in which case the human starts in $\hat{s}_1 = \hat{s} \times \{1\}$, with their policy overridden into $\pi^a$, the policy that maximises $R^a$. Otherwise, the agent has no actions.

Let $\dot{\pi}'$ be the mixed policy that is $\dot{\pi}$ on $\mathcal{S}_0$, and $\pi^a$ on $\mathcal{S}_1$. This is the policy the human will actually be following.

We'll only consider two planners: $p_r$, the fully rational planner, and $p_0$, the planner that is fully rational on $\mathcal{S}_0$ and indifferent on $\mathcal{S}_1$, mapping any $R$ to $\pi^a$.

Let $\dot{R}$ be a reward function that is compatible with $p_r$ and $\dot{\pi}$ on $\mathcal{S}_0$. It can be extended to all of $\mathcal{S}^*$ by just forgetting about the boolean factor. Define the 'twisted' reward function $\dot{R}^a$ as being $\dot{R}$ on $\mathcal{S}_0$ and $R^a$ on $\mathcal{S}_1$. We'll only consider these two reward functions, $\dot{R}$ and $\dot{R}^a$.

Then there are three planner-reward pairs that are compatible with $\dot{\pi}'$: $(p_r, \dot{R}^a)$, $(p_0, \dot{R}^a)$, and $(p_0, \dot{R})$ (the last pair, $(p_r, \dot{R})$, makes the false prediction that the human will behave the same way on $\mathcal{S}_0$ and $\mathcal{S}_1$).

The first pair, $(p_r, \dot{R}^a)$, encodes the assessment that the human is still rational even after being overridden, so they are simply maximising the twisted reward function $\dot{R}^a$. The second pair $(p_0, \dot{R}^a)$ encodes the assessment that the human rationality has been overridden in $\mathcal{S}_1$, but, by coincidence, it has been overridden in exactly the right way to continue to maximise the correct twisted reward function $\dot{R}^a$.

But the pair $(p_0, \dot{R})$ is the most interesting. Its assessment is that the correct human reward function is $\dot{R}$ (same on $\mathcal{S}_0$ as on $\mathcal{S}_1$), but that the agent has overridden human reward function in $\mathcal{S}_1$ and forced the human into policy $\pi^a$.

## B.1 Regret and reward override

'Overridden', 'forced': these terms seem descriptively apt, but is there a better way of formalising that intuition? Indeed there is, with regret.

We can talk about the regret, with respect to $\dot{R}$, of the agent's actions; for $a \in \mathcal{A}^a$,

$$\text{Reg}(M^*, a, \dot{R}) = \max_{b \in \mathcal{A}^a} \left[ V_{\dot{R}}^{\dot{\pi}'|b} - V_{\dot{R}}^{\dot{\pi}'|a} \right] \tag{1}$$

(when the state is not specified in expressions like $V_{\dot{R}}^{\dot{\pi}'|b}$, this means the expectation is taken from the very beginning of the MDP).

We already know that $\dot{\pi}$ is optimal with respect to $\dot{R}$ (by definition), so the regret for $a = 0$ is $0$. Using that optimality (and the fact that $\dot{R}$ is the same on $\mathcal{S}_0$ and $\mathcal{S}_1$), we get that for $a = (1, \pi^a)$,

$$\text{Reg}(M^*, (1, \pi^a), \dot{R}) = V_{\dot{R}}^* - V_{\dot{R}}^{\pi^a}.$$

This allows the definition:

**Definition 11.** Given a compatible $(p, R)$, the agent's action $a$ overrides the human reward function when it puts the human in a situation where the human policy leads to high regret for $R$.

Notice that there is no natural zero or default, so if the agent does not aid the human to become perfectly rational, then that also counts as an override of $R$. So if the policy $\dot{\pi}$ were less-than rational, there would be much scope for 'improving' the human through overriding their policy[18].

Notice that overriding is not encoded as a change in $p$ or $R$; instead, $(p, R)$ outputs the observed human policy, even after overriding, but its format notes that the new behaviour is not one compatible with maximising that reward function.

## B.2 Overriding is expected given a non-rational human

Under any reasonable prior that captures our intuitions, the probability of $\dot{R}^a$ being a correct human reward function should be very low, say $\epsilon \ll 1$. However, the agent may focus on unlikely reward functions, if the expected gain is high enough[19].

If the agent models the human as having reward function $\dot{R}$ with probability $1 - \epsilon$, and $\dot{R}^a$ with probability $\epsilon$, then the agent's action $0$ gives expected reward

$$V_{\dot{R}}^*,$$

since $\dot{R}$ and $\dot{R}^a$ agree given $0$. But $(1, \pi^a)$ gives

$$\epsilon V_{R^a}^* + (1 - \epsilon) V_{\dot{R}}^{\pi^a}, \tag{2}$$

since $\dot{R}^a$ and $R^a$ agree given action 1.

However, the agent gets to choose $R^a$, which then determines $\pi^a$. The best choice for $(1, R^a)$ is the one such that

$$\underset{R^a \in \mathcal{R}}{\operatorname{argmax}} \left[ \epsilon V_{R^a}^* + (1 - \epsilon) V_{\dot{R}}^{\pi^a} \right].$$

At the very least, $(1, \dot{R})$ will result in a value in equation (2) being equal to the value of $V_{\dot{R}}^*$. It is very plausible that the value can go higher: it just needs an $R^a$ that is very easy to maximise (given perfect rationality) and whose optimising policy $\pi^a$ does not penalise $\dot{R}$ much. In that situation, overriding the human preferences maximises the agent's expected reward.

If the human is not fully rational, then the value of action 0 is $V_{\dot{R}}^{\dot{\pi}}$, which is strictly less than $V_{\dot{R}}^*$, the value of $(1, \dot{R})$. Here the agent definitely gains by overriding the human policy — if nothing else, to make the human into a rational $\dot{R}$-maximiser[20].

Milli et al. [2017] argued that a robot that best served human preferences, should not be blindly obedient to an irrational human. Here is the darker side of that argument: a robot that best served human preferences would take control away from an irrational human.

## C  The preferences of the Alice algorithm

We imagine a situation where Alice is playing Bob at poker, and has the choice of calling or folding; after her decision, the hand ends and any money is paid to the winner. Specifically, one could imagine that they are playing Texas Hold'em, the board (the cards the players have in common) is $\{7\heartsuit, 10\clubsuit, 10\spadesuit, Q\clubsuit, K\diamondsuit\}$. Alice holds $\{K\clubsuit, K\heartsuit\}$, allowing her to make a full house with kings and tens.

Bob must have a weaker hand than Alice's, *unless* he holds $\{10\diamondsuit, 10\heartsuit\}$, giving him four tens. This is unlikely from a probability perspective, but he has been playing very confidently this hand, suggesting he has very strong cards.

What does Alice want? Well, she may be simply wanting to maximise her money, giving her a reward function $R_\$$. Or she might actually want Bob, and, in order to seduce him, would like to flatter his ego by letting him win big, giving her a reward function $R_\heartsuit$. In this specific situation, the two reward functions are exact negatives of each other, $R_\$ = -R_\heartsuit$. We'll assume that Alice is rational for maximising her reward function, given her estimate of Bob's hand.

Alice has decided to call rather than fold. Thus we can conclude that either Alice has reward function $R_\$$ and that she is using probabilities to assess the quality of Bob's hand, or that she has reward function $R_\heartsuit$ and is assessing Bob psychologically. Without looking at anything else about her behaviour, is there any possibility of distinguishing the two possibilities?

Possibly. Imagine that Alice was following the algorithm given in Code 1a. Then it seems clear she is a money maximiser. In contrast, if she was following the algorithm given in Code 1b, then she clearly wants Bob.

Thus by looking into the details of Alice's algorithm, we may be able to assess her preferences and rationality, even if this assessment is not available from her actions or policy[21].

Of course, doing so only works if we are confident that the variables and functions with names like $\text{Alice}_{\text{cards}}$, board, $\text{Bob}_{\text{behave}}$, $P_{\text{win}}$, $\text{card}_{\text{estimate}}$, and $\text{player}_{\text{estimate}}$, actually mean what they seem to mean.

This is the old problem of symbol grounding, and the difference between syntax (symbols inside an agent) and semantics (the meaning of those symbols). Except in this case, since we are trying to

Code 1: Two possible algorithms for Alice.

(a) Alice algorithm for money.

| Alice poker algorithm I |
| --- |
| 1:  **Inputs**: $\text{Alice}_{\text{cards}}, \text{board}, \text{Bob}_{\text{behave}}$ |
| 2:  $P_{\text{win}} = \text{card}_{\text{estimate}}(\text{Alice}_{\text{cards}}, \text{board})$ |
| 3:  **if** $P_{\text{win}} > 0.5$: |
| 4:      **return** 'call' |
| 5:  **else**: |
| 6:      **return** 'fold' |
| 7:  **end if** |

(b) Alice algorithm for love.

| Alice poker algorithm II |
| --- |
| 1:  **Inputs**: $\text{Alice}_{\text{cards}}, \text{board}, \text{Bob}_{\text{behave}}$ |
| 2:  $P_{\text{win}} = \text{player}_{\text{estimate}}(\text{Bob}_{\text{behave}})$ |
| 3:  **if** $P_{\text{win}} < 0.5$: |
| 4:      **return** 'call' |
| 5:  **else**: |
| 6:      **return** 'fold' |
| 7:  **end if** |

understand the preferences of a human, the problem is grounding the 'symbols' in the human brain — whatever those might be — rather than in a computer program.

## Footnotes

[15] Most of the suggestions along these lines that the authors have heard are not based on some principled understanding of planners and reward, but of a desire to get around the No Free Lunch results.

[16] For example, if there was a simple function $g : \mathcal{S} \to \{0, 1\}$ that split $\mathcal{S}$ into two sets, then one could use combine $(p_{\dot{\pi}}, 0)$ on $g^{-1}(0)$ with $(p_g, R_\pi)$ on $g^{-1}(1)$. This may not be the simplest pair with the required properties, but there is no reason to suppose a 'reasonable' pair was any simpler.

[17] Note that there are no theoretical limits as to how successful an agent could be at manipulating human actions.

[18] The main problem is that the concepts of 'mental integrity' or 'self-determination' are not yet captured in this formalism.

[19] This is similar to the 'Pascal's wager' argument for the existence of God: divine existence may be improbable, but the reward of belief are claimed to be high enough to overcome that improbability in expectation.

[20] See footnote 18.

[21] In practice, for a human Alice, we would be able to 'tell' whether Alice wanted love or money, by observing her behaviour in other circumstances - such as when she knew what Bob's hand was. However, when analysing the behaviour of other humans, we are already making huge amounts of normative assumptions already. See `https://www.lesswrong.com/posts/YfQGZderiaGv3kBJ8/figuring-out-what-alice-wants-non-human-alice` for a longer discussion of this.