[Reviews · NeurIPS 2018]

Reviewer 1



Summary: The paper addresses the inverse reinforcement learning problem and the ambiguity that exists in that ill-posed problem. The authors claim that one cannot learn only a reward to explain human behavior but should learn both the reward and the planner at the same time. In that case, they show that many couple (planner, reward) can explain the observed human behavior (or preferences) including a planner that optimizes the reward that is exactly the opposite of the true reward. The contribution is two-fold. First, they provide a bound for the worst case regret of a policy. Second they show that rewards that are compatible with the expert policy and have the lower complexity can be very far away from the actual reward optimized by the expert policy. It includes the opposite reward. Thus regularization would not help to reduce ambiguity. Quality: I think the paper makes good points in demonstrating that IRL is a complex problem and that current algorithms are not taking good care of some obvious issues. Decoupling the planner and the reward and show the ambiguity comes (at least for some part of it) from this complementarity may be one way to address these issues. But the authors are mainly arguing that human behavior is complex and that human preferences cannot be explained rationally. Yet, these arguments are not supported by any experimental clues. There is no human involved in a task in this paper and the state of the art section only gives a very short overview of how this research is related to research in human factors etc. I found this paper very high level, with strong claims but with very little actual facts supporting the claims. Clarity: Actually, I'm not sure about what is the main claim of the paper. Usually, IRL is used to mimic the behavior of some agent that is supposed to perform well at a task. Here, it looks that the authors are more interested in how the learnt reward could be used to explain the goal and preferences of the expert. Yet, the bound on the regret is more about performance than explainability. All in all, I think the authors could make good points but should make their goal clearer. Originality: I'm not familiar with the literature on human factors and psychology so I'm not sure I can judge the originality in a general context. I know some work in economics about rationality of agents that could be worth to be cited and discussed. In the machine learning community, I think this work is somewhat original. Significance: As the authors don't provide much solutions to the issues they describe, I'm not sure this paper can have a significant impact on the community. Ambiguity in the reward function is a well known problem and we know that the regularization techniques we use don't guarantee to recover the true reward. Yet, they still produce close enough performances and come with some guarantees too. I think this paper should be improved to place the work in a clearer context to have a better impact.

Reviewer 2



Summary: The paper shows (1) that it is impossible to uniquely decompose a policy into a planning algorithm and reward function, and (2) that even a reasonable simplicity prior on the set of compatible decompositions cannot distinguish between the true decomposition and others that lead to high regret. Together, these two theoretical results demonstrate that in order to accurately infer a human's reward function from demonstrations, we need to make normative assumptions about human planning that cannot be derived from observations of the human policy. Most prior work in inverse reinforcement learning models suboptimal behavior using noisy actions, e.g., Boltzmann rationality in maximum entropy IRL. A few recent papers propose extensions that model systematic biases in human planning, like false beliefs, temporal inconsistency, risk sensitivity, and internal dynamics model misspecification. This paper gives a theoretical motivation for further work on modeling human irrationality in order to accurately infer preferences from actions. Strengths: The problem setup is quite general, which enables the results to apply broadly to the various inverse reinforcement learning and Bayesian inverse planning frameworks. Each theorem is accompanied by intuitive explanations; for example, the connection to Hume's is-ought problem in Section 4.1, and the characterization of the function B in lines 239-240. Weaknesses: I was a little confused by the introduction of the function B in line 255, but I finally understood its usefulness when I saw it vanish in the last step of line 265 to establish the complexity lower bound! A few minor corrections: - Line 91 should start with instead of - Line 253 has some typos - Line 255 should contain (-pa, -Rr) instead of (pa, Rr) Overall, the paper has excellent quality, originality, and significance. Clarity could be improved a bit by addressing typos.

Reviewer 3



This paper presents a novel and important insight in value alignment: that the No Free Lunch theorem for inferring human preferences cannot be alleviated using simplicity priors. It also claims to introduce this theorem, which was however already proved in Everitt et al, 2017 (as Theorem 11). The paper states that "A No Free Lunch result analogous to theirs applies in our case" (section 4.1), but it's not clear to me in what sense this result is analogous rather than identical (since the theorem statement and assumptions are exactly the same). This paper presents a different proof for the No Free Lunch theorem - please make clear how this proof is an improvement on the proof in the Everitt et al paper. The presentation of the main insight (that simplicity priors don't work) has some issues as well. The argument relies on a key assumption that B(p-dot, R-dot) is complex even given human policy pi-dot (in section 5.2 on informal complexity results), which is made precise as l_L(pi-dot, B(p-dot, R-dot)) > l_L(pi-dot) + 5C (in section 5.4 on formal complexity results). This is a strong assumption that is not justified in the paper, and the coefficient 5 for the constant C is chosen in a seemingly contrived way to make the proof of Theorem 8 work out. The larger the constant C, the stronger this assumption is, and the constant C could be large, since it needs to simultaneously satisfy all the inequalities in Theorem 7. For the same reason, the result of Theorem 7 is not very impressive: the three pairs of interest are among the 2^(-K) N > 2^(l_L(pi-dot) + 4C) simplest pairs in the language L, which is exponential in C and thus is very large if C is large. The paper should provide some upper bounds on the value of C to make these results more meaningful. I expect the central claim of the paper probably still holds, and the informal argument in section 5.2 makes sense, but it needs more rigorous support. The paper is clearly written and well-motivated, though a bit unpolished: - There are some incorrect theorem labels: Corollary 4 refers to theorems 4.1 and 4.2, which should be theorems 1 and 3. - Some terms are not defined before they are used, e.g. "simple function". - The conclusion section ends abruptly and seems unfinished. The paper includes a thorough literature review and makes clear how this work relates to previous works and how this approach is more general than previous approaches to the problem. It opens up new avenues for research into normative assumptions for deducing human preferences from behavior. UPDATE after reading author feedback: - The explanation of the assumption that B(p-dot, R-dot) is complex given the human policy seems convincing, so I'm happy to increase my score. - I still think there is not much novelty in the NFL theorem. The planning function in this setting is analogous to the reward corruption function in the Everitt et al (2017) paper. The Everitt paper's NFL theorem shows that if we don't assume anything about the reward corruption function, the worst-case regret of any policy is high. This paper's NFL theorem analogously shows that if we don't assume anything about the planning function, the worst-case regret of any policy is high. The statement in the author feedback that the setting in the Everitt paper is "unrelated to reward inference" does not seem right, since that paper is about the agent's ability to infer the true reward function. Thus, I don't think it took much work to transfer the NFL theorem to the preference inference setting. Overall, I would recommend that the authors put less emphasis on the NFL theorem and more emphasis on the "Simplicity does not help" section, since that is the main novel and impactful contribution of the paper.